# Development of ANN-Based Warpage Prediction Model for FCCSP via Subdomain Sampling and Taguchi Hyperparameter Optimization

**DOI:** 10.3390/mi14071325

**Published:** 2023-06-28

**Authors:** Hsien-Chie Cheng, Chia-Lin Ma, Yang-Lun Liu

**Affiliations:** Department of Aerospace and Systems Engineering, Feng Chia University, Taichung 407, Taiwan; charliema962@gmail.com (C.-L.M.); yanglliu@fcu.edu.tw (Y.-L.L.)

**Keywords:** flip-chip chip-scale package, warpage prediction model, artificial neural network, process modeling, hyperparameter optimization, sampling strategy, viscoelasticity

## Abstract

This study aims to establish an accurate prediction model using artificial neural networks (ANNs) to effectively and efficiently predict the process-induced warpage of a flip-chip chip-scale package (FCCSP). To enhance model performance, a novel subdomain-based sampling strategy and Taguchi hyperparameter optimization are proposed in the ANN algorithm. To simulate the warpage behavior the FCCSP during fabrication, a process modeling approach is proposed, where the viscoelastic behavior of the epoxy molding compound is included, in which the viscoelastic properties are determined using dynamic mechanical measurement. In addition, the temperature-dependent thermal-mechanical properties of the materials in the FCCSP are assessed through thermal-mechanical analysis and dynamic mechanical analysis. The modeled warpage results are verified by the warpage measurement. Next, warpage parametric analysis is performed to identify the key factors most affecting warpage behavior for use in the construction of the warpage prediction model. Moreover, the advantages of the proposed sampling and hyperparameter tuning approaches are proved by comparing with other existing models, and the validity of the developed ANN-based deep learning warpage prediction model is demonstrated through a validation dataset.

## 1. Introduction

Nowadays, semiconductor manufacturing technologies have advanced rapidly, driven by the demand for smaller, faster, and more reliable electronic devices. However, this advancement has also brought up a significant technical challenge, in which the physical limit of transistor scaling creates enormous difficulties in continuing on the path of Moore’s Law [1]. In the post-Moore era, the concept of “More than Moore” based on heterogeneous integration using new packaging technologies [2,3,4,5,6,7,8] is becoming more critical and demanding. Flip-chip chip-scale packaging (FCCSP) possesses the capacity of a high I/O count, miniaturization, and great electrical performance, and thus becomes one of the promising packaging solutions for realizing heterogeneous system integration.

Despite the fact that FCCSP has been one of the mainstream packaging technologies today (see, e.g., [6,7,8]), there are still several technical issues that need to be addressed, such as yield, reliability, thermal performance, and warpage. Among them, the warpage induced during the manufacturing process is particularly important as it can cause various process problems in subsequent process steps, such as handling, registration, and alignment, eventually resulting in yield and throughput losses [9,10]. It is, thus, essential to have a thorough understanding of its warpage behavior in the initial design stage. In the literature, few studies have been reported on the characterization and management of the warpage behavior of FCCSP during fabrication through theoretical analysis, such as finite element analysis (FEA), and experiments [8]. As compared to experimental approaches, theoretical analysis can be not only more efficient and cost-effective, but also capable of giving better insight into the physical mechanisms.

To solve the aforementioned challenges and even lessen the prediction uncertainty and modeling error made by less experienced engineers, researchers start seeking the integration of simulation and machine learning (see, e.g., [11,12,13,14,15,16]). To date, due to the rapid advance of computer technologies and machine learning algorithms, it has evolved into a critical tool for addressing a wide range of real-world issues, with applications covering medical diagnosis, transportation, space exploration, defense systems and various engineering fields. Deep learning is a branch of machine learning which incorporates artificial deep learning neural networks (NNs). These NNs are composed of a number of neurons, each of which performs a simple mathematical function of the inputs. By assembling these layers, deep learning models have the ability to learn complex information and features from raw data, such as images, audio, text, and sensor readings. This has allowed the development of a deep learning-based prediction model for timely and effective predictive analysis of very complex systems. The simulation-based deep learning prediction models have been extensively applied in advanced microelectronic packaging for a quick and accurate assessment of their thermal-mechanical performance, such as thermal performance [12,13] and reliability [14,15,16]. For example, Law et al. [12] developed a deep learning model for the prediction of the thermal performance of quad flat no-lead (QFN) packages using an ANN. Subbarayan et al. [14] applied an artificial NN (ANN) algorithm to build up a reliability prediction model for a ball grid array (BGA) package. Yuan et al. [15] applied an ANN-based simulation framework to investigate the solder joint reliability of a wafer-level chip-scale package, where the initial parameters of the ANN model, namely, the weights and bias, were obtained using a genetic algorithm (GA). Hsiao and Chiang [16] combined FEA together with random forest (RF) to explore the solder joint reliability of wafer-level packaging subjected to thermal cycling.

In addition to conventional gradient-based back-propagation (BP) approaches, evolutionary algorithms (EAs), such as GA, evolutionary strategy (ES), and particle swarm optimization (PSO), have been extensively applied to network topology design and connection weight adaption [17,18,19,20], mainly because of their advantages over the conventional approaches’, such as conceptual simplicity and flexibility, capability to solve problems without any human expertise, and higher probability to reach a global optimum. White and Ligomenides [17] proposed a two-stage approach to explore the network topology and connection weights of an NN model by combining a GA and a BP approach. The underlying idea behind this approach is that if the GA was unable to obtain an appropriate network solution, the BP approach with an MP algorithm was further performed to locally explore the optimal weights using the calculated connection weights from the GA as initial values. A similar approach can also be found in Ding et al. [18]. Juang [19] introduced an evolutionary recurrent network for a temporal sequence production problem using an evolutionary learning algorithm based on a hybrid of GA and particle swarm optimization (PSO). Ahmadizar et al. [20] developed an ANN model using an evolutionary-based algorithm that integrates grammatical evolution (GE) for the network topology design and GA for better weight adaptation.

The NN prediction model performance can be alternatively improved through hyperparameter tuning [21,22,23,24,25]. Hyperparameters are crucial for the performance of a machine learning model because they control the architecture of a neural network. Well-tuned hyperparameters can also prevent the model from overfitting or underfitting (see, e.g., [26]). In the literature, the hyperparameters were mostly tuned using trial-and-error parametric analysis (one factor at a time) [21], grid search [22], and random search [23]. The former two approaches are either unable to account for the interaction effect of hyperparameters, or computationally expensive, especially for models with a large number of hyperparameters and a huge search space. Random search could be a more efficient and cost-effective approach; however, theoretically, it is less probable to find the optimal hyperparameter setting. EAs, such as GAs [24,25], are a feasible alternative to determine the best set of hyperparameters. Even though Erpolat Taşabat and Aydin [25] found that for hyperparameter optimization, GAs can be more efficient in computation than grid search, the heuristic algorithms may fail to converge to an optimal or even good result due to their premature convergence in nature, and, in addition, they are computationally cost-ineffective due to their poor convergence. Consequently, a more effective and cost-effective hyperparameter optimization approach is preferred and needed.

According to the above literature survey, there are still very limited studies on the development of the warpage prediction model for electronic packaging, not to mention FCCSP. Thus, this work attempts to develop a prediction model using a proposed FEA-based ANN approach to facilitate an effective and quick estimate of the process-induced warpage behavior of FCCSP for use in subsequent fabrication process design. In order to upgrade model prediction accuracy and training performance, an ANN algorithm integrating a novel subdomain-based sampling strategy and Taguchi hyperparameter optimization is proposed for prediction model design and training. To simulate the fabrication process, an FEA-based process modeling approach is proposed, which takes into account the viscoelastic behavior of the epoxy molding compound (EMC) and the temperature-dependence of the thermal-mechanical properties of the materials in FCCSP. For the validation of the proposed process modeling approach, the warpage simulated results are compared against the warpage measurement data. Moreover, warpage parametric analysis is performed to characterize the crucial factors that mainly influence the warpage behavior. These characterized crucial factors are utilized for the ANN prediction model’s construction. The benefits of the proposed sampling and hyperparameter tuning techniques are shown by comparison to other existing approaches. Furthermore, the feasibility of the developed warpage prediction model is evaluated using the validation dataset.

## 2. Structure and Fabrication Process of FCCSP

The FCCSP assembly under investigation is depicted in Figure 1, which mainly consists of a silicon die, an EMC, copper pillar bumps (CPBs), and a coreless substrate. The coreless substrate used in this study is a three-layer 168 µm thick embedded trace substrate (ETS), which comprises two solder mask (SM) protective layers, two prepreg (PP) dielectric layers, and three metal (Cu) layers. The schematic diagram of the cross section of the coreless substrate is shown in Figure 2. In this investigation, three FCCSP test vehicles (TV) with different geometric dimensions are discussed. Taking TV1 as an example, the die is 8.6 mm in length, 8.2 mm in width, and 200 μm thick. The die is connected on the coreless substrate using copper pillar bumps, and then the electronic assembly is fully covered by an EMC material with a size of 15 mm (length) 15 mm (width) 430 μm (thickness). Compared to TV1, the EMC thickness of TV2 and of TV3 is 450 μm. The die thicknesses for TV1, TV2 and TV3 are 200 μm, 200 μm and 175 μm, respectively. The detailed dimensions of TV1, TV2, and TV3 are shown in Table 1. Figure 3 illustrates the fabrication process of the FCCSPs, which includes two major process steps, namely die bonding process (steps 0–3) and mold cure process (steps 3–6). The die bonding process starts by mounting the silicon die on the coreless substrate using copper pillar bumps. The silicon die is aligned with the bond pads on the substrate and then heated to 260 °C to activate the solder bumps and form an electrical and mechanical connection. Once the die-bonding process is completed, a liquid-type EMC is used to encapsulate the silicon die through the mold cure process with a mold cure temperature of 175 °C. The mold cure process helps provide additional electrical insulation and environmental protection.

## 3. Theoretical Model of ANN

ANNs, data processing models, are designed to mimic the human nervous system [12]. The architecture and behavior of ANNs are inspired by the biological NNs in human brains, which process information in a parallel and distributed manner. A typical ANN model mainly comprises three layers, namely, the input layer, hidden layer, and output layer, as shown in Figure 4. The input layer, i.e., the first layer of an ANN model, is primarily responsible for receiving the external inputs. The hidden layers, the intermediate layers or the neural layers between the input layer and the output layer, manage the ANN’s data processing and computation. Increasing the hidden layers enhances the capability of mimicking a more complex and nonlinear features and behaviors, meanwhile raising the computational complexity and effort, and potentially causing overfitting and poor prediction performance. The output layer, the last layer of an ANN model, is in charge of providing predictions based on the computations performed in the hidden layers. The links connecting neurons in an ANN model are termed connection weights, which are to be solved through optimization. Figure 4 illustrates the process of passing information through an NN having two inputs (*x*_1_, *x*_2_) and outputs (*o*_1_, *o*_2_), one hidden layer with three neurons (*z*_1_, *z*_2_, *z*_3_) inside, where *w* is the weight, *b* the bias of the layer, and σ the activation function of the layer. The goal of an ANN model is to modify the weights through optimization or learning process to minimize the discrepancy of the ANN outputs and the target data. Additionally, the setting of hyperparameters of an ANN model, including optimizer, number of hidden layers and neurons, activation function, learning rate, and batch size, is critical to the prediction model’s performance. The most commonly used hyperparameter optimization methods include trial-and-error parametric analysis [21], grid search [22], random search [23] and EAs such as GA [24,25]. However, these methods hold various drawbacks (see Introduction). Consequently, a more effective and cost-effective approach using the Taguchi method is proposed to determine the optimal hyperparameter setting for constructing the best-fitted prediction model.

## 4. Process Modeling

An FEA-based process modeling approach that integrates the ANSYS element death/birth technique and nonlinear FEA is introduced for effectively evaluating the warpage of the FCCSP during the fabrication process. Considering its symmetry, a quarter-symmetric FEA model of the FCCSP is adopted, where a symmetric boundary condition is imposed on these symmetric planes, i.e., the nodal displacements normal to the symmetric planes are zero. In addition, to avoid rigid body motion, the displacement of the bottom node on the intersecting line of these two symmetric planes is constrained in the z-direction. The FEA model of the FCCSP is primarily composed of a coreless substrate, an EMC, Cu pillar bumps, and a silicon die, as shown in Figure 5, together with the imposed boundary conditions. Hexahedral solid elements in ANSYS, i.e., solid 185, are adopted. Table 2 lists the number of nodes and solid elements of the FEA models associated with TV1, TV2, and TV3. The Young’s modulus (E) and coefficient of thermal expansion (CTE) of the EMC, prepreg, Sn-Ag-Cu(SAC)305 solder, and solder mask are characterized using a thermal-mechanical analyzer (TMA) (TA Instruments, New Castle, DE, USA) and a dynamic mechanical analyzer (DMA) (TA Instruments, New Castle, DE, USA), and the results are displayed in Figure 6. Except for the EMC, which is assumed to be a linearly viscoelastic material, they are considered to be linearly elastic, isotropic, and temperature-dependent. In addition, the CTEs and Young’s moduli of the silicon die and Cu are 2.8 ppm/°C and 160 GPa, and 16.3 ppm/°C and 121 GPa, respectively. According to the fabrication process displayed in Figure 3, the process modeling primarily involves the die bonding process (steps 0–3) and mold cure process (steps 3–6). At step 0, the silicon die, solder layer of the CPB, and EMC are deactivated. At step 1, i.e., heating to the die bonding temperature (260 °C), the solder layer and silicon die are activated to form a mechanical connection between the silicon die and the coreless substrate. At step 4, i.e., heating to the mold cure temperature (175 °C), the EMC is activated to simulate a fully cured EMC.

EMC materials play a significant role in the thermal-mechanical behavior of electronic packaging [9]. Typically, EMC materials reveal temperature-, time- and strain-rate-dependent viscoelastic behaviors (see, e.g., [10]), such as creep, stress relaxation, and even hysteresis behavior. The viscoelastic relaxation behavior is generally depicted by a generalized Maxwell model, comprising multiple Maxwell elements and an independent spring connected in parallel. This generalized Maxwell model is well approximated by a Prony series representation for fitting measured relaxation data,
(1)E(t)=E∞+∑i=1mEiexp−tτi,
wherein E(t) denotes the relaxation modulus of the entire model, $E_{i}$ the relaxation modulus of the *i*th Maxwell element, $E_{\infty }$ the long-term fully relaxed modulus, t the time, $\tau_{i}$ the relaxation time, and m the total number of Maxwell elements. Based on the following relationship between the unrelaxed modulus $E_{0}$ and $E_{\infty }$,
(2)$$E_{0}=E_{\infty }+\sum_{i=1}^{m}E_{i},$$
the Prony series representation of the generalized Maxwell model (Equation (1)) can be rewritten as
(3)$$E(t)=E_{0}\left[\beta_{\infty }+\sum_{i=1}^{m}\beta_{i}\exp\left(-\frac{t}{\tau_{i}}\right)\right],$$
where $\beta_{i}$ represents $E_{i}/E_{0}$.

The time and temperature dependence of the mechanical properties of a viscoelastic material can be correlated using the time–temperature superposition principle (TTSP) [10]. More specifically, the TTSP suggests that a relaxation curve of a viscoelastic material at a specific temperature can be employed as a reference for further characterizing the relaxation curves at other temperatures by conducting a horizontal translation of the reference relaxation curve in the logarithmic time domain. The temperature translation factor $\lambda_{T}$ is normally approximated using an empirical relationship, the so-called Williams–Landel–Ferry (WLF) equation,
(4)$$\log_{10}\lambda_{T}=\frac{-\kappa_{1}\left(T-T_{r}\right)}{\kappa_{2}+\left(T-T_{r}\right)},$$

In Equation (4), $\kappa_{1}$ and $\kappa_{2}$ are the curve fit coefficients, and $T_{r}$ the reference temperature.

The master curve of the relaxation modulus at a reference temperature can be constructed by translating the measured frequency-dependent storage moduli at multiple temperatures along the time axis with temperature translation factors $\lambda_{T}$. Based on the relaxation modulus at different isothermal temperatures under 1% applied strains [10], the constructed reference master curve at the glass transition temperature of the EMC is shown in Figure 7 and the fitted coefficients ($\beta_{i}$,$\tau_{i}$) of the Prony series model with 21 terms are given in Table 3. Furthermore, the fitted coefficients $\kappa_{1}$ and $\kappa_{2}$ of the WLF model for the characterized translation factors as a function of temperature are 74.7 and 313.9, respectively.

## 5. Results and Discussion

### 5.1. Characterization of Process-Induced Warpage of FCCSP

The process-dependent warpage evolution of these three FCCSP test vehicles (TV1, TV2 and TV3) during the fabrication process is calculated, and displayed in Figure 8. The stress-free temperature of the substrate is set 145 °C to model the initial warpage of the substrate, i.e., about 67 µm ± 15 µm. The results show a significant rise in warpage after the die bonding process step. This dramatic increase in warpage suggests that the process temperature plays a significant role in the warpage. Furthermore, the process-induced warpage of the FCCSP is significantly reduced after the mold cure process, mainly owing to the EMC’s ability to reduce the CTE mismatch between the substrate and die. This implies the capability of the EMC for suppressing the warpage. The FEA results are compared with the warpage measurement data obtained using Shadow Moiré, as shown in Table 4. Note that the warpage is measured on the bottom surface of the substrate at room temperature after the mold cure process, i.e., step 6, using a Shadow Moiré measurement technique (Akrometrix TherMoire AXP 2.0, Atlanta, GA, USA). In addition, the warpage is defined as the discrepancy between the maximum and the minimum of the z-direction deformation. The viscoelastic effect of the EMC on the process-induced warpage of the FCCSP is also examined. It is found that the warpage result after the mold cure process obtained from the process modeling approach considering the EMC viscoelastic effect shows a much more consistency with the measurement data than that without considering the effect, indicating that the EMC viscoelastic effect is essential for the prediction of the process-induced warpage. The modeled and measured warpage contour plots of the FCCSP after the mold cure process are presented in Figure 9, where the FCCSP would deform in a convex shape. Moreover, the minimal warpage takes place at the center of the FCCSP while the maximal warpage takes place at the four corners. Evidently, these two warpage contours also agree well with each other. The close agreement in warpage between the simulation (with the EMC viscoelastic effect) and the measurement clearly proves the effectiveness of the proposed process modeling approach in warpage prediction.

### 5.2. Identification of Key Factors Affecting Process-Induced Warpage

The influences of some geometric and material factors on the process-induced warpage of the FCCSP are investigated through parametric analysis using the validated FEA-based process modeling approach. The considered geometric and material factors are the side length and thickness of the die, side length of the package, thickness of the EMC, CTE of the EMC and substrate, and Young’s modulus (E) of the substrate. It should be noted that the variation of the side length of the package would correspondingly change the side length of the substrate and EMC while keeping the dimension of the other components unchanged. These design factors are nominally varied by ±15% from their nominal values. Note that the width and length of this FCCSP package are identical, and those of the silicon die are very similar. Since there is a very comparable parametric analysis result with respect to the width and length of the package and silicon die, they are simply replaced by the “side length” of the package and silicon die for better clarity and conciseness of presentation. The parametric results of the effects of the side length of the silicon die and package, and the effect of the thickness of the die and EMC are presented in Figure 10a. The process-induced warpage is found to increase with an increasing die side length and a decreasing package side length. This is mainly because as the die’s side length goes up, the mechanical stresses because of the CTE mismatch between the top layer (the composite layer of the EMC and die) and the bottom substrate become more pronounced, thereby resulting in an increased warpage. On the other hand, the decrease in package side length would also reduce the size of the substrate and EMC, which as the die size remains unchanged, would increase the proportion of the silicon die in the top layer, and resultingly enhance the CTE mismatch between the composite top layer and the substrate. Furthermore, an increased warpage can be also observed with an increasing die thickness and a decreasing EMC thickness, primarily due to the growth in the CTE mismatch between the composite top layer and the bottom substrate as a result of the increased proportion of the silicon die in the top layer. Figure 10b summarizes the parametric results of the influences of the Young’s modulus and CTE of the substrate and the CTE of the EMC. It demonstrates that a decrease in the CTE and Young’s modulus of the substrate would diminish the process-induced warpage, while an increase in the CTE of the EMC would lessen it. The explanations can also be found above.

### 5.3. Establishment of Training/Test and Validation Datasets

Based on the results of the parametric analysis, the degree of influence of these seven factors on the warpage behavior of the FCCSP after the fabrication process is ranked from the highest to the lowest as follows and as also listed in Table 5: EMC thickness, substrate CTE, EMC CTE, die side length, die thickness, substrate Young’s modulus and package side length. Out of them, the top six highest-influence factors are chosen to establish the ANN prediction model for the process-induced warpage. They are considered as the input parameters in the ANN input layer, and are also used to establish the training/test dataset. A variation range of ±20% is considered for these input parameters. For the establishment of the training/testing dataset, several sampling strategies are available. Two of the most widely used ones are global structured (GS) sampling and global random (GR) sampling. GS sampling is a well-established factorial design-based sampling strategy in which a full factorial design (FFD) of design of experiment (DOE) is utilized for the entire design domain (the design region of the factors) to create the sample data. This method exploits all the combinations of factors at all levels. It is known for its ability to achieve an even distribution of sample data across the entire design domain, which in turn could give a better assessment of the interactions among factors. Because the number of sample data increases with an increase in the number of factors, the number of sample data could be very large if the amount of input features and levels are excessive, probably leading to a high computational cost [21]. In addition, this strategy needs to reconstruct the sampling datasets using an FFD of DOE when more data are needed to obtain a more accurate prediction, thereby being less flexible in additional data generation. In contrast, the GR is a sampling strategy that randomly selects a subset of the population (sample data) from the entire population (the entire designed region of the factors). This is a very simple and straightforward method, as compared to the GS sampling strategy, since it has no need of prior knowledge about the sampling population. In addition, because of the use of randomization, this strategy would better avoid sampling and selection biases and enjoy high flexibility in generating additional data if needed. Nevertheless, this method may result in an uneven data distribution, which is not beneficial to a thorough and effective evaluation of the interactions among factors.

To take into account the flexibility and feasibility of additional data generation without the need of re-establishing the sampling dataset, and achieve an even data distribution, this study proposes a subdomain random (SR) sampling strategy to construct the datasets by partitioning the whole design domain into multiple subdomains, in which a random generation of sample data is made. The schematic diagrams of the GS, GR, and SR sampling strategies are shown in Figure 11. In addition, two of their combinations (the so-called hybrid approaches), namely, GS combined with GR (hereafter termed the GSGR strategy), and GS combined with SR (hereafter termed the GSSR strategy), are also proposed. The training/testing dataset is constructed using these five sampling strategies and their results in terms of model performance are compared to each other. Four different sample data sets are considered for the GS, GR and SR strategies, i.e., 216, 324, 540, and 810. For the GSGR and GSSR hybrid sampling strategies, three additional different sample data sets, namely, those with 108, 324, and 594 samples, are generated by GR and SR, respectively, and they are further combined with the 216-sample data set generated by the GS to form three sample data sets, i.e., data sets of sizes 324, 540, and 810. For GS, the corresponding factors and levels together with the total number of sample data used in the training/test phase are presented in Table 6. The same total number of sample data are created using GR. For SR, an increase in the number of subdomains would enhance the sampling and modeling complexities. To reduce the number of subdomains, any two design factors are grouped together into one cluster, and each of them is further divided into two regions. For this six-factor design, a total of eight subdomains can be formed. Beside the training/test dataset, a validation dataset with sixty-four sample data is established using GS to verify the prediction accuracy of the trained ANN warpage prediction model. The factors and levels used in the validation phase for GS are listed in Table 7.

### 5.4. Hyperparameter Optimization Using Taguchi Method

The design of a machine learning prediction model consists in seeking hyperparameter optimization. In addition, the optimal values of the hyperparameters are highly problem-dependent. In this investigation, the initial/untuned hyperparameter values, namely, optimizer (Adam) [27], activation function (ReLU) [15], the number of hidden layers (three) [28], and the estimate of the upper limit of the number of neurons in each layer [29], are specified based on the literature results. Note that the size of the training dataset is directly related to the number of neurons applied in each layer. The learning rate is set according to the default value of the Keras Adam optimizer, and the fold number (K) of the K-fold cross-validation method is set according to the size of the test dataset, which is 10~15% of the training data. The GS sampling strategy is used to form the training/test dataset. Moreover, the untuned hyperparameter values used for constructing the ANN models with four different training/test datasets (i.e., 216-, 324-, 540- and 810-sample data) are listed in Table 8. Min–max normalization is applied to scale these input features to a fixed range such that each feature has a comparable weight for the feature learner. The mean square errors (MSEs) of the predictions of the trained ANN models on the test data for these four training/test datasets are shown in Table 9. It is clear to see that these MSE values are considerable, indicating that there is a significant degree of discrepancy between the predictions and calculations, and, also, there is a great room to improve the ANN prediction models.

To improve the model’s performance, the Taguchi method is applied to determine the optimal hyperparameter values. For the six-factor, three-level experimental design problem, one two-level factor and seven three-level factors are employed in Taguchi’s orthogonal array (OA), as shown in Table 10, to seek the optimal combinations of hyperparameters and levels. According to the previous literature reports, three hidden layers are found to effectively reduce the calculation time, and, meanwhile, provide good prediction accuracy [28]; the optimizer Adam could achieve good results with high efficiency on most neural network architectures [27]. Thus, they are implemented in the ANN model. The considered hyperparameters for optimization are activation function (A), batch size (B), learning rate (C), number of neurons in the first hidden layer (D), number of neurons in the second hidden layer (E), number of neurons in the third hidden layer (F), and two dummy factors. For the hyperparameter optimization, the training/test dataset with 324 sample data is used. The hyperparameters and their levels used in the Taguchi experimental design are presented in Table 11. The MSE of the predictions of the trained ANN model on the test dataset is considered as the objective of the Taguchi experimental design. The the-smaller-the-better criterion is used for the minimization of the MSE. The signal-to-noise (S/N) ratios of all the experimental runs in the OA are calculated. The mean S/N ratio for each level of control factors is summarized in the S/N response graph shown in Figure 12. The response graph is utilized to identify the most significant hyperparameters and their optimal level set for achieving an improved performance of the trained ANN warpage prediction model. From this response graph, it is found that the optimal level set of these hyperparameters is A1, B3, C3, D3, E3, and F3, i.e., exponential linear unit (ELU) activation function, a batch size of 30, a learning rate of 0.005, and 11 neurons for all three hidden layers.

### 5.5. Performance Characterization and Comparison of the Trained ANN Models on Validation Dataset

With the optimal set of hyperparameters, four ANN models are trained again using these four different training/test datasets (i.e., 216, 324, 540 and 810 sample data) generated with the GS sampling strategy. The MSEs and their standard deviations on the test data associated with the four training/test datasets are characterized and described in Table 12. For comparison, the corresponding results of the trained ANN models with the untuned hyperparameter setting are also presented in this table. Noteworthy is that for each training/test dataset, the same training/test data are utilized for both the untuned and tuned hyperparameter settings. Evidently, this demonstrates that both the prediction accuracy (MSEs) and precision (standard deviations) of the trained ANN models with the tuned hyperparameter setting are exceptionally improved for all these four training/test datasets, suggesting that the present hyperparameter optimization using the Taguchi method is an effective and feasible means of enhancing the performance of the ANN prediction model. With the same optimal hyperparameter setting, the ANN models are also trained on these four different training/test datasets generated by the other four sampling strategies (namely, GR, SR, GSGR, GSSR). In total, there are twenty trained ANN prediction models in accordance with the four training/test datasets and five sampling strategies.

After the ANN models were suitably trained, the aforementioned validation dataset with sixty-four sample data generated with the GS sampling strategy was further used to assess and compare the performance of these twenty trained ANN warpage prediction models. The corresponding prediction performances of the ANN models with the tuned hyperparameter settings are summarized in Table 13 and Figure 13, in terms of the difference in the average warpage with standard deviation and the maximum warpage between the calculations and predictions. The following facts can be observed from this table. First of all, it is clear to see that a larger number of training/test data tends to yield a better prediction result in terms of both the average warpage and maximum warpage for all these five sampling strategies. This result is aligned with the literature findings, such as Panigrahy et al. [21]. Then, among these five sampling strategies, without a doubt, the GS strategy would have the best prediction performance, irrespective of the training/test datasets applied, due to its even data distribution, while the GR would obtain the worst prediction results due to an uneven data distribution. It is, however, pointed out that even though the GS can obtain the best prediction results, it needs to completely reconstruct the whole sampling dataset using an FFD of DOE when more sample data are in demand for better prediction accuracy, thus requiring a much higher computational and modeling effort in data generation. Next, it is interesting to see that the proposed SR sampling strategy tends to demonstrate a superior prediction capability than the GR, and even the GSGR, especially in the average warpage difference between the calculations and predictions, despite having a poorer performance than the GS. Apart from that, the proposed GSSR hybrid sampling strategy outperforms not only the GR and GSGR but also the proposed SR. Finally, the GS and the proposed GSSR provide a very comparable prediction performance, but the latter is comparatively much more flexible in producing more sample data and is also more computationally cost-effective.

## 6. Conclusions

This study successfully establishes an ANN-based deep learning warpage prediction model using a novel subdomain-based sampling technique and Taguchi hyperparameter optimization to facilitate the process-induced warpage prediction and design of the FCCSP in the initial design stage. To characterize the process-dependent warpage behavior of the FCCSP, an FEA-based process modeling approach that takes into account the viscoelastic behavior of the EMC material. The effectiveness of the proposed process modeling approach is extensively demonstrated by comparing the simulated results with the measured data. The validated process modeling approach is subsequently applied in both the parametric analysis for exploring the key factors most affecting the process-induced warpage behavior, and the construction of the warpage prediction model using the ANN. The superiority of the proposed sampling and hyperparameter tuning techniques is extensively justified by comparing with other existing models, and the applicability of the constructed warpage prediction model is well confirmed using the validation dataset. Several essential conclusions are deduced below.

1.The proposed process modeling approach turns out to be very effective in the assessment of the process-dependent warpage evolution of the FCCSP, where after the mold cure process, the FCCSP would deform in a simple convex shape, where the minimal warpage takes place at the center of the FCCSP, and the maximal warpage at the four corners.2.Process temperature is found to play a significant role in the process-dependent warpage, and, in addition, the die bonding process step would induce significant warpage while the mold cure process step would suppress the warpage, primarily due to the CTE effect of the EMC.3.The viscoelastic behavior of the EMC is crucial for an accurate estimate of the process-induced warpage behavior of the FCCSP.4.Parametric analysis shows that an increasing side length and thickness of the silicon die and the CTE and Young’s modulus of the substrate, and a decreasing side length of the package and thickness and CTE of the EMC would enlarge the process-induced warpage.5.Taguchi hyperparameter optimization suggests that the optimal hyperparameters are the exponential linear unit (ELU) activation function, a batch size of 30, a learning rate of 0.005, and 11 neurons for all these three hidden layers. It turns out that the Taguchi method can be a very effective and feasible way to augment the model’s performance.6.The prediction results can be improved with a larger number of training/test data tends, which is in good match with the literature findings.7.Among these five sampling strategies (GS, GR, SR, GSGR, GSSR), GS tends to provide the best prediction accuracy because of its even data distribution, and, in contrast, GR reveals the worst because of likely producing an uneven distribution of sample data. It is worth mentioning that in spite of having the best prediction results, GS demands a much greater computational and modeling effort to totally reconstruct the entire sampling dataset when additional sample data are needed.8.Even though having an inferior performance than the GS, the proposed SR sampling strategy surpasses both the GR and GSGR. More importantly, the proposed GSSR outperforms not only GR and GSGR but also the proposed SR. It is also interesting to find that even though both GS and the proposed GSSR can give an equivalent prediction performance, GSSR is the preferred choice because of its greater flexibility in generation of additional sample data, thereby being less computationally demanding.

## Figures and Tables

**Figure 1 micromachines-14-01325-f001:**
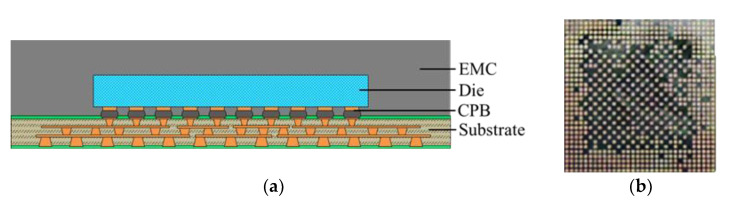
The FCCSP assembly. (**a**) Cross-sectional view. (**b**) Prototype.

**Figure 2 micromachines-14-01325-f002:**
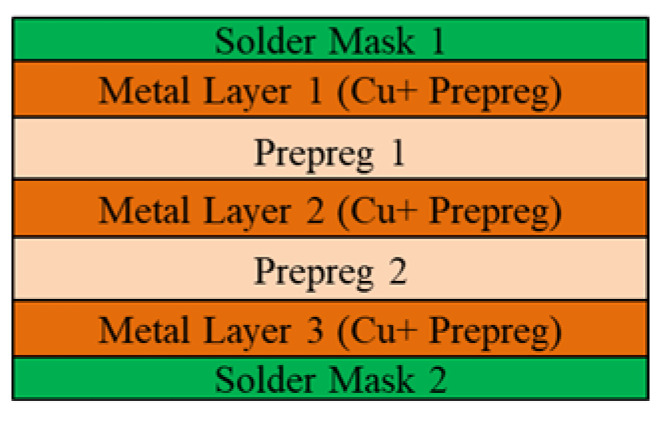
Schematic cross-sectional view of coreless substrate.

**Figure 3 micromachines-14-01325-f003:**
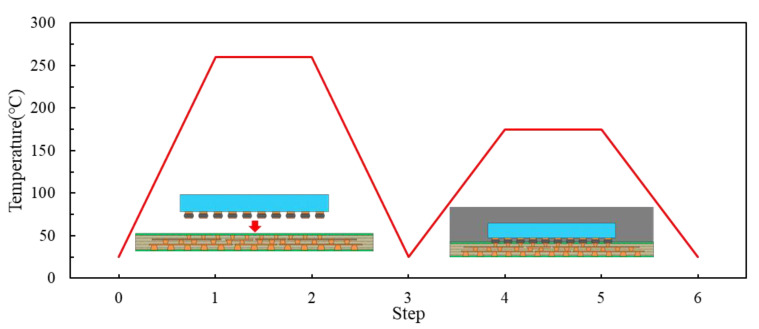
Fabrication process steps of FCCSP.

**Figure 4 micromachines-14-01325-f004:**
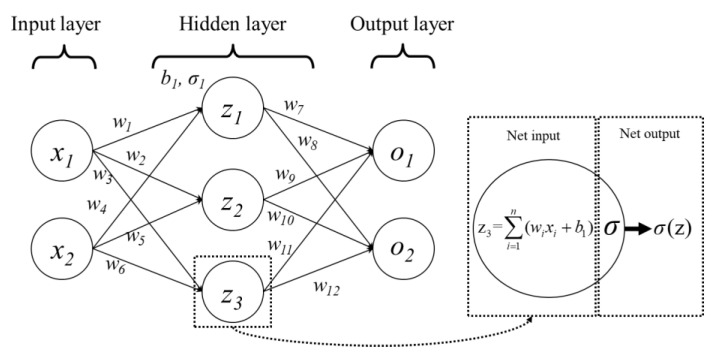
Schematic diagram of ANN.

**Figure 5 micromachines-14-01325-f005:**
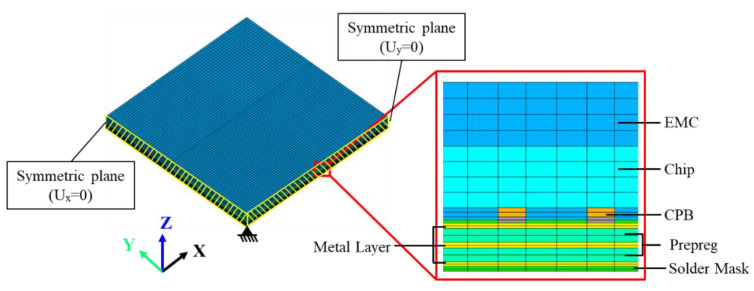
3D FEA model of the FCCSP.

**Figure 6 micromachines-14-01325-f006:**
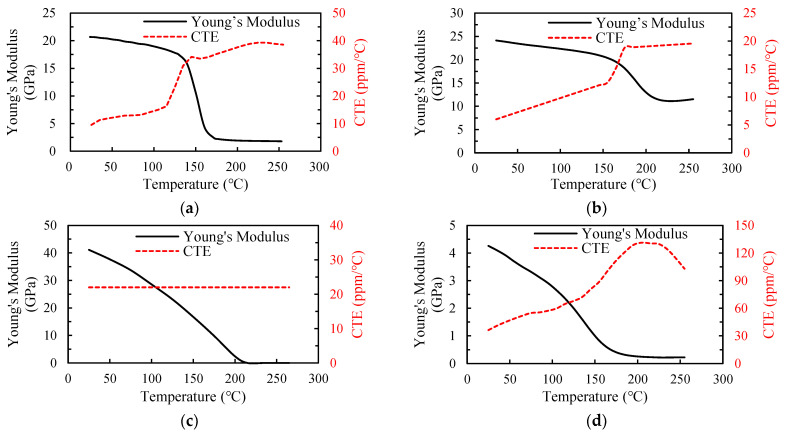
Thermal-mechanical properties of materials: (**a**) EMC, (**b**) prepreg, (**c**) SAC solder and (**d**) solder mask.

**Figure 7 micromachines-14-01325-f007:**
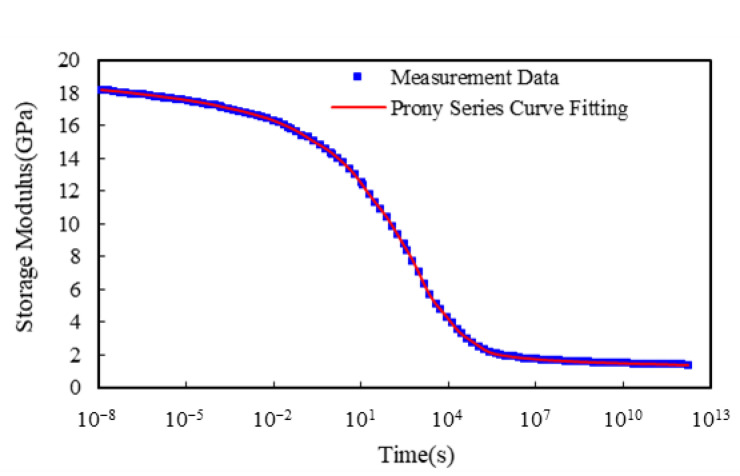
Established reference master curve of relaxation modulus and its Prony series curve.

**Figure 8 micromachines-14-01325-f008:**
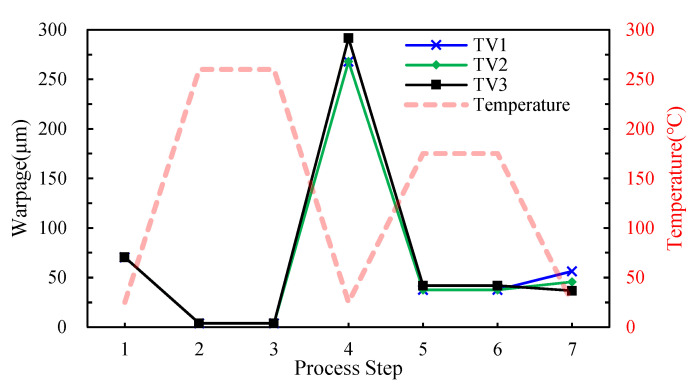
Warpage variations for the TV1, TV2, and TV3 during fabrication process.

**Figure 9 micromachines-14-01325-f009:**
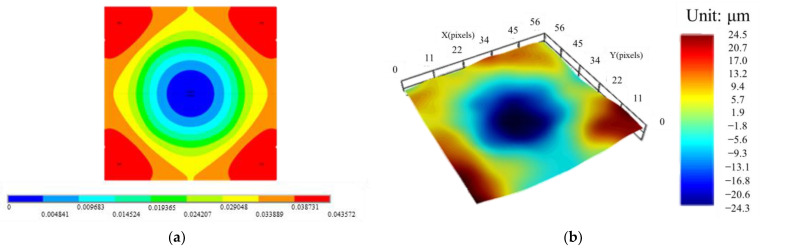
The (**a**) simulated and (**b**) measured warpage contour plots.

**Figure 10 micromachines-14-01325-f010:**
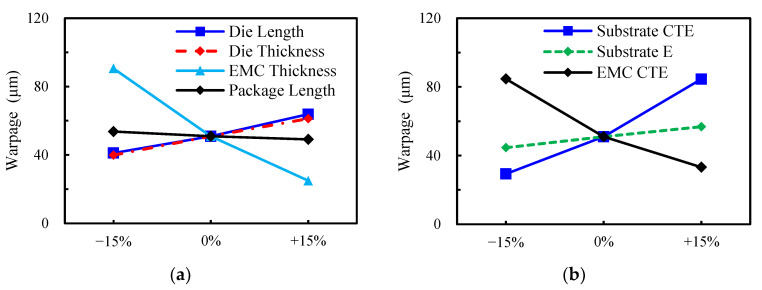
Effects of components’ (**a**) geometric parameters (length, width and thickness) and (**b**) material constants (CTE and Young’s modulus).

**Figure 11 micromachines-14-01325-f011:**
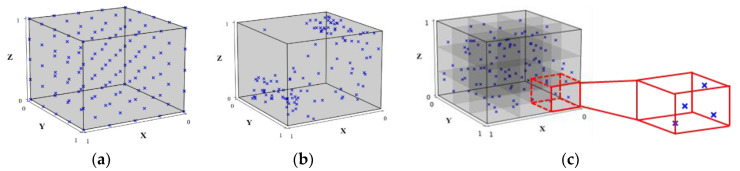
Schematic of the three sampling strategies: (**a**) GS, (**b**) GR, and (**c**) SR.

**Figure 12 micromachines-14-01325-f012:**
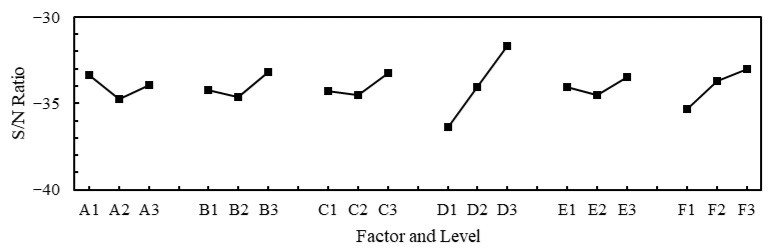
Effect of hyperparameters on S/N ratio.

**Figure 13 micromachines-14-01325-f013:**
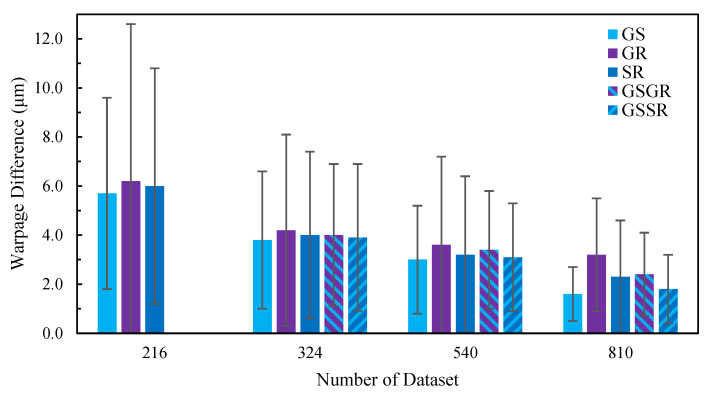
Influences of sampling strategies on prediction performance of the ANN models.

**Table 1 micromachines-14-01325-t001:** Dimension of FCCSP TV1, TV2, and TV3 (Unit: mm).

Structure	TV1		TV2		TV3	
	Thickness	Length × Width	Thickness	Length × Width	Thickness	Length × Width
EMC	430	15.0 × 15.0	450	15.0 × 15.0	450	15.0 × 15.0
Die	200	8.6 × 8.2	200	8.6 × 8.2	175	8.6 × 8.2
CPB	40	0.03 × 0.06	40	0.03 × 0.06	40	0.03 × 0.06
Solder Mask 1	10	15.0 × 15.0	10	15.0 × 15.0	10	15.0 × 15.0
Metal Layer 1	18	15.0 × 15.0	18	15.0 × 15.0	18	15.0 × 15.0
Prepreg 1	45	15.0 × 15.0	45	15.0 × 15.0	45	15.0 × 15.0
Metal Layer 2	18	15.0 × 15.0	18	15.0 × 15.0	18	15.0 × 15.0
Prepreg 2	45	15.0 × 15.0	45	15.0 × 15.0	45	15.0 × 15.0
Metal Layer 3	16	15.0 × 15.0	16	15.0 × 15.0	16	15.0 × 15.0
Solder Mask 2	16	15.0 × 15.0	16	15.0 × 15.0	16	15.0 × 15.0

**Table 2 micromachines-14-01325-t002:** Number of nodes and elements of the 3D FEA model.

	TV1	TV2	TV3
Nodes	131,400	136,656	136,656
Elements	122,475	127,800	127,800

**Table 3 micromachines-14-01325-t003:** Fitted Prony series coefficients.

*i*	$\tau_{i}$	$\beta_{i}$	*i*	$\tau_{i}$	$\beta_{i}$
1	1.0 × 10^−8^	0.006318	12	1.0 × 10^3^	0.211667
2	1.0 × 10^−7^	0.010797	13	1.0 × 10^4^	0.125250
3	1.0 × 10^−6^	0.011782	14	1.0 × 10^5^	0.074790
4	1.0 × 10^−5^	0.013151	15	1.0 × 10^6^	0.014854
5	1.0 × 10^−4^	0.019278	16	1.0 × 10^7^	0.008881
6	1.0 × 10^−3^	0.021791	17	1.0 × 10^8^	0.005064
7	1.0 × 10^−2^	0.029155	18	1.0 × 10^9^	0.004296
8	1.0 × 10^−1^	0.054915	19	1.0 × 10^10^	0.002483
9	1.0 × 10^0^	0.055408	20	1.0 × 10^11^	0.002877
10	1.0 × 10^1^	0.124230	21	1.0 × 10^12^	0.002496
11	1.0 × 10^2^	0.124722			

**Table 4 micromachines-14-01325-t004:** The measured and simulated warpages (unit: μm).

Test Vehicle		TV1	TV2	TV3
Measured	Average	49.4	41.2	38.0
	Warpage range	(42.0~58.0)	(35.0~49.0)	(34.0~42.0)
Simulated	(W/O Viscoelastic)	30.6	22.0	20.0
Simulated	(W/Viscoelastic)	53.6	43.6	34.9

**Table 5 micromachines-14-01325-t005:** Ranking of factors in terms of degree of influence.

Factor	Degree of Influence	Rank
EMC Thickness	128.7%	1
Substrate CTE	108.3%	2
EMC CTE	100.9%	3
Die length	44.5%	4
Die thickness	41.9%	5
Substrate E	23.9%	6
Package length	9.0%	7

**Table 6 micromachines-14-01325-t006:** Factors and levels corresponding to the four training/test datasets for the GR sampling strategy.

	Dataset			
	216	324	540	810
Factor	Level Value			
Die side length (mm)	6.4/9.6	6.4/8.0/9.6	6.4/8.0/9.6	6.4/8.0/9.6
Die thickness (mm)	0.16/0.24	0.16/0.24	0.16/0.24	0.16/0.20/0.24
EMC thickness (mm)	0.36/0.45/0.54	0.36/0.45/0.54	0.36/0.41/0.45/0.50/0.54	0.36/0.41/0.45/0.50/0.54
EMC CTE	−20%/20%	−20%/0%, 20%	−20%/0%/20%	−20%/0%/20%
Substrate CTE	−20%/0%/20%	−20%/0%, 20%	−20%/0%/20%	−20%/0%/20%
Substrate E	−20%/20%	−20%/20%	−20%/20%	−20%/20%

**Table 7 micromachines-14-01325-t007:** Factors and levels used in the validation phase.

Factor	Level Value
Die length (mm)	7.0/9.3
Die thickness (mm)	0.18/0.24
EMC thickness (mm)	0.38/0.44
EMC CTE	−18%/15%
Substrate CTE	−18%/19%
Substrate E	−18%/18%
Total data	64

**Table 8 micromachines-14-01325-t008:** Untuned hyperparameter values used for training ANN models under four different training datasets.

Hyperparameter	Training/Test Dataset			
	216	324	540	810
Optimizer	Adam	Adam	Adam	Adam
K-Fold number (K)	10	10	10	10
Neural number of hidden layers	(5, 5, 5)	(8, 8, 8)	(11, 11, 11)	(14, 14, 14)
Activation function	ReLU	ReLU	ReLU	ReLU
Learning rate	0.001	0.001	0.001	0.001

**Table 9 micromachines-14-01325-t009:** Prediction MSEs of the trained ANN models on the test data.

Training/Test Dataset	216	324	540	810
MSE	77.8 ± 25.1	46.9 ± 11.4	28.8 ± 5.3	20.9 ± 8.1

**Table 10 micromachines-14-01325-t010:** $L_{18}(2^{1}\times 3^{7})$ OA and variable assignments.

EXP	Factor							
	A	B	C	D	E	F	Dummy	
1	1	1	1	1	1	1	1	1
2	1	2	2	2	2	2	1	2
3	1	3	3	3	3	3	1	3
4	2	1	1	2	2	3	1	3
5	2	2	2	3	3	1	1	1
6	2	3	3	1	1	2	1	2
7	3	1	2	1	3	2	1	3
8	3	2	3	2	1	3	1	1
9	3	3	1	3	2	1	1	2
10	1	1	3	3	2	2	2	1
11	1	2	1	1	3	3	2	2
12	1	3	2	2	1	1	2	3
13	2	1	2	3	1	3	2	2
14	2	2	3	1	2	1	2	3
15	2	3	1	2	3	2	2	1
16	3	1	3	2	3	1	2	2
17	3	2	1	3	1	2	2	3
18	3	3	2	1	2	3	2	1

**Table 11 micromachines-14-01325-t011:** Hyperparameters and levels considered for Taguchi hyperparameter optimization.

Factorial Levels and Their Values		Level 1	Level 2	Level 3
A.	Activation function	ELU	ReLU	Leaky ReLU
B.	Batch size	10	20	30
C.	Learning rate	0.0005	0.001	0.005
D.	Neural number in hidden layer 1	5	8	11
E.	Neural number in hidden layer 2	5	8	11
F.	Neural number in hidden layer 3	5	8	11

**Table 12 micromachines-14-01325-t012:** MSEs on test data predicted by the ANN models with the untuned and tuned hyperparameter (HP) settings (unit: μm).

Sampling Strategy	HP Setting	Training/Test Datasets			
		216	324	540	810
GS	Untuned	77.8 ± 25.1	46.9 ± 11.4	28.8 ± 5.3	20.9 ± 8.1
	Tuned	23.5 ± 8.1	11.2 ± 3.3	10.6 ± 2.6	8.0 ± 2.9
GR	Untuned	75.1 ± 56.0	37.5 ± 30.1	20.9 ± 20.5	9.8 ± 2.2
	Tuned	14.5 ± 9.1	7.7 ± 4.3	7.8 ± 3.3	6.5 ± 1.6
SR	Untuned	32.8 ± 44.2	18.0 ± 11.0	8.3 ± 3.3	6.9 ± 2.2
	Tuned	9.9 ± 4.0	8.4 ± 2.8	6.9 ± 3.1	5.5 ± 2.8
GSGR	Untuned	-	39.5 ± 10.2	26.7 ± 5.9	20.0 ± 6.9
	Tuned	-	18.9 ± 6.8	8.6 ± 2.8	7.3 ± 2.2
GSSR	Untuned	-	41.7 ± 8.4	27.5 ± 10.9	21.5 ± 11.3
	Tuned	-	19.5 ± 7.5	14.1 ± 5.3	8.4 ± 4.6

**Table 13 micromachines-14-01325-t013:** Warpage prediction performance of the ANN models with the tuned hyperparameter setting (unit: μm).

Sampling Strategy	Training/Test Datasets							
	216		324		540		810	
	Avg.	Max	Avg.	Max	Avg.	Max	Avg.	Max
GS	5.7 ± 3.9	15.8	3.8 ± 2.8	10.4	3.0 ± 2.2	9.8	1.6 ± 1.1	4.3
GR	6.2 ± 6.4	38.8	4.2 ± 3.9	20.4	3.6 ± 3.6	20.2	3.2 ± 2.3	10.0
SR	6.0 ± 4.8	18.6	4.0 ± 3.4	17.2	3.2 ± 3.2	20.2	2.3 ± 2.3	11.7
GSGR	-	-	4.0 ± 2.9	14.2	3.4 ± 2.4	12.2	2.4 ± 1.7	7.1
GSSR	-	-	3.9 ± 3.0	13.4	3.1 ± 2.2	9.8	1.8 ±1.4	6.5

## Data Availability

Data sharing is not applicable.

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
