# Peer review of "Development of ANN-Based Warpage Prediction Model for FCCSP via Subdomain Sampling and Taguchi Hyperparameter Optimization"

_micromachines, 2023, doi:10.3390/mi14071325_

Round 1

Reviewer 1 Report

Dear Authors,

Before a final decision can be made regarding the acceptance of your manuscript, I believe some aspects need further clarity and precision. Here are my suggestions:

  1. While Table 1 provides dimensions of the FCCSP, it would be beneficial if more specific details concerning the geometry and dimensions of each layer are included. For instance, you might consider illustrating a figure with each layer's dimensions.
  2. The source references for Figures 4 and 6 should be provided if these figures have been cited from other sources.
  3. The boundary conditions of the simulation model should be described in a more comprehensive manner, either graphically or textually.
  4. The setup and implementation of dead-birth elements in your model need to be described more clearly.
  5. Please provide a detailed methodology for how warpage data collection was conducted, specifying aspects such as time, location, etc.
  6. Your manuscript could greatly benefit from the addition and detailed description of the simulation model and meshing model.
  7. The quality of Figures 3 and 9 should be improved for better clarity and understanding.
  8. It would be beneficial to provide a detailed description of the in-line warpage measurement process and how it compares with the simulation.

Sincerely,

Reviewer 2 Report

- Each statement should be followed by a reference that confirms that information.
- Table 1 should be presented more transparently and clearly. Length, width, thickness should be indicated in the box in words. Solder mask must contain the numbers 1 or 2, as well as prepreg. It is necessary to use the same type of description as in Figure 1a) - in words, not abbreviations. EMC is epoxy molding compound - so it should be written in both Figure 1 and Table 1.
- A space should be placed after "±" through the text.
- In Figure 9, a better image resolution should be used.
- Figure 10 is not clearly shown. What should the x-axis represent, and where is the width parameter of the component shown.

Round 2

Reviewer 1 Report

The last version is good for publishing.

Sincerely yours,